# Gold Nanoparticles and Plant Pathogens: An Overview and Prospective for Biosensing in Forestry

**DOI:** 10.3390/s22031259

**Published:** 2022-02-07

**Authors:** Prabir Kumar Kulabhusan, Anugrah Tripathi, Krishna Kant

**Affiliations:** 1Institute for Global Food Security, School of Biological Sciences, Queen’s University, Belfast BT9 5DL, UK; p.kulabhusan@qub.ac.uk; 2Monitoring and Evolution Division, Directorate of Research, Indian Council of Forestry Research and Education, Dehradun 248006, India; anugrahtrip@gmail.com; 3Departamento de Química Física, Campus Universitario, CINBIO Universidade de Vigo, 36310 Vigo, Spain

**Keywords:** plant pathogen biosensing, plant disease, forest diseases, gold nanoparticles

## Abstract

Plant diseases and their diagnoses are currently one of the global challenges and cause significant impact to the economy of farmers and industries depending on plant-based products. Plant pathogens such as viruses, bacteria, fungi, and pollution caused by the nanomaterial, as well as other important elements of pollution, are the main reason for the loss of plants in agriculture and in forest ecosystems. Presently, various techniques are used to detect pathogens in trees, which includes DNA-based techniques, as well as other microscopy based identification and detection. However, these methodologies require complex instruments and time. Lately, nanomaterial-based new biosensing systems for early detection of diseases, with specificity and sensitivity, are developed and applied. This review highlights the nanomaterial-based biosensing methods of disease detection. Precise and time effective identification of plant pathogens will help to reduce losses in agriculture and forestry. This review focuses on various plant diseases and the requirements for a reliable, fast, and cost-effective testing method, as well as new biosensing technologies for the detection of diseases of field plants in forests at early stages of their growth.

## 1. Introduction

In recent years, innovatory applications of nanomaterials have been described in several fields of science and technology. The outstanding properties owned by nanomaterials have encouraged scientists to connect their vast potential in physical, chemical, and biological sciences [1]. In contrast to their bulk equivalents, the nanoscale dimensions provide special optical, electronic, physical, or mechanical properties to nanomaterials. Metal based, non-metal, and metal-based oxide nanomaterials are widely used today, and a vast range of applications such as drug delivery, medical devices and diagnostics, material research, optics, and imaging are attributed to these. Amongst several types of nanomaterials that include carbon nanotubes, graphene, and other metal-based nanoparticles, gold nanoparticles (AuNPs) have meaningfully been used in miscellaneous applications involving nanodevices and products [2]. The remarkable properties of AuNPs, such as a high surface area-to-volume ratio, biocompatibility, and an ease in surface functionalization with recognition probes (e.g., antibodies, ap-tamers, peptides) make them amenable for various detection platforms. The localized surface plasmon resonance (LSPR) properties of AuNPs can be monitored to visualize the dispersion and aggregation states. Depending upon the size of the AuNPs and the aggregation of the particles can lead to the color changes from pink to violet to pale blue [3]. This concept has been used in several diagnostic platforms and colorimetric detection of various analytes. Apart from this, AuNPs have been utilized for bioanalysis during SPR transduction. This feature is based on the change of the dielectric constant of the surface plasmons of the AuNPs, and the analytes can be detected based upon the changes in angle, intensity, or the phase of the reflected light [4]. Furthermore, the AuNPs have the capability to transfer electrons between the electrode and electroactive biological species. This fundamental transfer has been employed for the redox enzyme biosensing, where the recognition probe, such as an enzyme, catalyze the oxidation and reduction of the analytes. Considering the multitude facets of the AuNPs, the application of AuNPs in various biosensors and detection modalities have been explored [5].

Use of AuNPs in biosensors is remarkable due to the distinctive colorimetric or optical detection possible with these nanoparticles. Biosensing of disease-causing pathogens in plants using nanomaterials is one such area where incredible developments have been completed [6]. Existence of diseases in plants is one of the foremost causes of production and yield losses in the agricultural sector [7]. Avoiding disease appearance and blowout has led to an enormous use of chemicals in the form of pesticides (fungicides and bactericides), which not only stands a large ecological encounter, but also increases the input costs of production. Both cultivation and post-cultivation (storage) diseases in crop plants are instigating huge losses to farmers and industries globally. Condition with forestry species, mainly woody perennials, is also worrying. Forests and tress-outside-forests (TOFs) play a pivotal role in the resilience and adaptation of our ecosystem under changing global environmental scenarios, mitigation of climatic variations, and carbon storage; however, the rising incidence of weather fluctuations and extremes due to accelerated anthropogenic interferences are showing increasingly negative impacts on forests due to the critical thresholds already being exceeded [8].

Tree diseases are the primary reason of productivity losses in the forest each year and it is estimated that the average total loss of timber due to disease-caused mortality and growth loss approximately equals the losses caused by all other stress factors in totality. Over the years, there have been several major pathogen outbreaks in the tress that have drastically affected several species and ecosystems negatively across the globe [9]. Diseases and pest attacks seriously affect the health of forests, as well as trees outside the forests (TOFs). Tree health refers to all integrated abiotic and biotic factors that affect the vigor and productivity of a tree, as expressed by various symptoms and types of damage. The health of forests can be assessed qualitatively by describing the symptoms or damage, or quantitatively through assessment of physiognomy [10]. Forest tree species across the globe are quite susceptible to disease, as well as insect-pest attacks on agricultural crops, which cause multifarious ecological and economic losses. The extent of damage by insects, pests, and disease depends on the stage at which they are infested and the causal insect/pathogens. Among the insect pests, trees are usually attacked by leaf eaters, defoliators, stem borers, sap suckers, gall formers, seed in-festers, etc. Diseases in trees are generally caused by fungi, bacteria, and viruses. Common diseases on trees, which are caused by fungi, are rust, leaf blight, leaf spot, powdery mildew, wilt, and root rot, etc. Bacterial cankers, bacterial leaf scorch, necrosis, crown gall, and chlorosis are common bacterial disease in trees [11].

The most widely used strategies to control pathogens in trees are severe pruning, roguing of infected branches, and the control of the insect vector via insecticide/fungicide applications. Both of these strategies may not be economically and ecologically viable, as the former requires extensive labour, and later affects our natural ecosystem and soil negatively. The effective management and mitigation measures of forest tree health problems depend upon their early and accurate detection, particularly initiated with the initial diagnosis and recognition of symptoms in the field. In addition to this, understanding its associated biology, recognizing the most susceptible stage, peak duration of activity, and its reciprocal relationship with the environment are also key steps to combat forest disease problems and avoid economic losses [7,12]. Creating awareness and capacity building for the forest fringe communities and other stakeholders are very crucial to show responses to health-related problems of the forests. Broadening of visual skills in the initial assessment of forest health is required to improve the early detection and timely management of issues [13]. In the changing scenario of silvicultural operations and management systems, the pest and disease control has acquired new horizons, which has become more complex and requires high throughput tools and techniques for early disease sensing.

In developing countries and countries in transition, severe pest outbreaks in forests may severely affect national economics, undermine local livelihoods associated with the adjoining forests, and threaten food security. The methods for detection of disease and its causal organism that are developed should be highly sensitive, specific, and provide rapid results. Therefore, nanotechnology-based biosensors have proved to be very effective in several agricultural crops. Fewer studies on the application of nano-sensors for forest tree disease sensing have been conducted but adopting these methods in large areas of forest affected by disease outbreaks will surely benefit to growers, farmers, and forest departments for enhancing forest productivity and carbon fluxes.

Disease detection approaches have largely been based on either direct or indirect detection methods. The conventional methods such as microscopic examination of pathogen through culture and colony count, immunological detection assays, and molecular methods like polymerase chain reaction are all direct detection methods where molecular, morphological, and serological properties of pathogens are directly targeted for detection. These are considered as the ‘gold standard’ but are quite challenging, require specific skills, dedicated instrumentation, sample preparation steps, and normally the time taken in obtaining results and its interpretation is longer [14] There are indirect methods wherein any physiological or morphological changes taking place in the infected plants are targeted for detection [7]. These methods involve techniques like optical imaging, thermography, gas chromatography, and mass spectrometry. All these methods are quite effective in detecting the causal organisms but have their own limitations and benefits. Deciding a suitable method depends upon the purpose, stage of the disease, location, ease of carrying out the experiment, and availability of test sample and costs. Rapid detection of plant diseases is beneficial in terms of the timely action to control the spread and minimize yield losses. Most of the direct and indirect methods of detection are difficult to implement in on-site settings and require more time in providing results. Hence the development of quick and simple sensing platforms or devices has been key to research areas in plant disease management.

Nanomaterial-based biosensors are being used today in a plethora of applications related to environmental monitoring and medical diagnostics. Their use in plant sciences or agriculture has witnessed a tremendous increase in recent years. A typical biosensor comprises of a bio-recognition element, a transducer, and a signal interpretation system. Gold nanoparticles and nanostructures have been used on a large scale in the immobilization of bio-recognition elements and provides for the exploitation of surface plasmon resonance effects in transducing signals. Use of nanoparticles have been found to increase the performance of biosensing devices in terms of limit-of-detection and sensitivity of the sensor [15,16]. However, the application of other nanomaterials, such as carbon nanotube, graphene, and other metal nanoparticles are studied but the majority of them are used to test their toxicity effect over the plant cells. The known mechanism of AuNPs and easy surface modification with biocompatibility made AuNPs the most suitable candidate for the sensing application. Through this review, the authors prefer to focus on various plant diseases which occur in the forest regions and in the domestic fields. The requirement of time for a reliable, fast, and cost-effective testing method for the detection of field plants in forests and plants is at the early stages of disease growth. The review also details about the use and advantages of nanomaterial for the new biosensing technologies and plant disease detection as presented in Figure 1.

## 2. Plant Pathogens and Plant Diseases

Forest trees possess long rotation periods and are significant in environmental and commercial prospective. It is estimated that forests provide up to 16% of the human population’s needs, and 18% of cattle needs [17]. Across the globe, natural forests and plantations are subject to encounter damage by several diseases caused by virus, bacteria, fungi, and other parasitic organisms [18]. Pathogens are the key component in ecosystems and play a key role in the diversity and distribution of trees in the forest ecosystem. Disease and pest attacks in trees are among one of the major stressors in forestry, and forests are exposed for multiple pathogens, pest invasion, and nutrient deficiencies. Tree pests and disease are the ultimate nuisance for the health and wealth of forests. Various bio-pathogens, particularly fungi, bacteria, virus, oomycetes, phytoplama, and plant parasitic nematodes cause disease in forest trees. Most tree diseases are caused by fungi. The highest diversity of fungal pathogens causing foliar and wood rot disease is ascomycetes and basidiomycetes. Different type of symptoms of fungal disease in plants are leaf spots, rusts, scorch, blotch, anthracnose, and needle blights. These fungal pathogens may be visible by small fruiting bodies distinctive of pathogenic species. The key examples of fungal pathogens on trees are Fusarium, *Ganoderma*, *Rhizctonia*, *Melampsora*, *Erysiphe*, and *Armillaria* etc. [17]. *Anthracnose* is a common disease on trees, specifically in deciduous trees, which causes darker and sunken lesions on flowers, leaves, stems, and fruits. Powdery mildew causes a white coating over the foliar part, which results to the distortion, yellowing, and, ultimately, the death of leaves. It is caused by a fungus; mostly found on plants and trees in shady and high humid areas. Rust is a fungal disease which affects leaves with yellow spots on the upper leaf surface of hardwood species. (Figure 2A,B).

Bamboo is an important forest species, due to its recurring and fast growth, extraordinary strength properties, and multidimensional uses. Bamboos are the target of several pathogens among which fungi are main group that cause culm diseases (rot of emerging culm rot, culm stain, culm spot), foliar diseases (spot, rust, and blight), rhizome diseases (rhizome rot), and sheath diseases (sheath rot and sheath spot) [19] (Figure 2C).

Common bacterial diseases in trees include shot hole, foliar blight, and wilt, mainly caused by *Erwinia*, *Xanthomonas*, and *Ralstonia*. Viruses are non-living, sub microscopic, obligate mesobiotic parasites that multiply intracellularly. These are transmitted through contact or through physical and biological vectors. Various viral diseases, such as bamboo mosaic potexvirus, elm mottle virus, oak mosaic virus, and prunus necrotic ringspot virus are caused in forest trees by viruses. The propagation of most of the forest trees from seeds is difficult due to less availability of seeds; therefore, the seed viability preparation of seedlings through seeds and plantlets from clonal cuttings has paramount importance for massive afforestation programs. As seedlings and clonal plantlets are propagated extensively under monoculture systems, either in the nursery field or inside mist chambers under high moisture and abundant nutrient supply, these promote proliferation of pathogens. Common root disease in the nursery plants includes the pre-emergence of blight/damping-off, post-emergence damping-off, vascular wilt disease, root rot disease, set rot of cuttings, and web blight. Common foliar diseases are leaf spots, leaf blights, leaf scorch, leaf rusts, and powdery mildews. Therefore, disease biosensing at the nursery stage through nano-sensors has a huge scope in the forestry sector. Asia and Europe have each encountered losses of approximately 5 million hectares of forests, due to major outbreaks of forest tree diseases from across the world, as listed below in Table 1.

## 3. Gold Nanoparticle Based Biosensors

AuNPs have been used as an important component in various biological and chemical sensing devices. The remarkable characteristics, such as higher surface area-to-volume ratio, tunable optical properties, and ease in synthesis methods, have led to their usage in several diagnostic platforms of plant, animal, or human diseases, biomarkers, and other chemical analytes of importance [20]. AuNPs can be fabricated via different methods, namely, chemical and photochemical reduction, seed-mediated growth, and green synthesis. The Turkevich–Frens method involving trisodium citrate-based reduction of gold salts and the Brust–Schiffrin method involving external thiol ligands for the reduction of gold salts are the most used reaction methods for synthesizing AuNPs [21,22].

### 3.1. Electrochemical Biosensors

Electrochemical based detection methods exhibit a high sensitivity, low-cost, rapidness, and a capability of miniaturization. The working principle of an electrochemical sensor is conducted using three components, including a working electrode (WE), counter electrode (CE), and reference electrode (RE). Glass electrodes (glassy carbon electrode) are used as the working electrode, whereas platinum wire and Ag/AgCl (saturated KCl) usually act as the CE and RE, respectively. The binding of the target analytes causes the changes of current, potential, and impedance of the sensor, which provides the measurable signal with different detection approaches. The detection techniques mainly include the electrochemiluminescence (ECL), electrochemical impedance spectroscopy (EIS), and voltammetry such as differential pulse voltammetry (DPV), cyclic voltammetry (CV), square wave voltammetry (SWV), and anodic stripping voltammetry (ASV). Furthermore, the surface area of the WE is one of the important factors to determine the sensitivity of the sensor. Considering the huge surface-area-to-volume ratio, ease in modification, and better signal amplification, the application of AuNPs in electrochemical sensor offers a great advantage [23]. The AuNPs enhance the conductivity by modifying the sensing surface and catalysing the chemical reactions. The immobilization of recognition probes such as antibodies, aptamers, and peptides on the surface of AuNPs provides a better affinity towards the target analytes. Furthermore, the AuNPs acts as the electrochemical indicators based on the redox reaction between Au^0^ and Au^3+^. In electrochemical biosensors, the AuNPs signals are generally detected by: (a) direct detection of oxidation signal of AuNPs without any treatment [24], (b) the electro-oxidation of AuNPs to gold ions using hydrochloric acid (HCl) [14,25], and (c) treatment of AuNPs in HBr/Br2 solutions [26]. Researchers demonstrated the binding event between the single-stranded DNA binding protein (SSB) of E. coli and complimentary single-stranded oligonucleotides conjugated to AuNPs. The resulting AuNPs-conjugated SSB was used as the hybridization label and the changes in the AuNPs oxidation signal was measured. The limit of detection (LOD) of the sensor was found to be 2.17 pM of the target DNA [27].

### 3.2. Enzyme Biosensors

Enzymes have a unique catalytic activity and specificity towards its target substrate, which results in the wide usage of the biomedical application. The immobilization of enzymes has received great attention due to their multiple or repeatable use for a particular purpose and improving their stability [28]. However, the high production costs and expensive separation techniques sometimes deter their usages. There are several enzyme immobilization methods already reported in the literatures, mostly including the covalent bonding, adsorption, cross-linking, entrapment, and self-assembled monolayers. Covalent bonding involves the sharing of electron pair between the enzyme and the platform on which the molecule is immobilized leading to high strength and multipotent attachment. The covalent interaction is usually performed by two steps: firstly, the activation of the substrate (Au nanoparticles) using linker molecules, such as carbodiimide or thiol, occurs. Secondly, the other end of the linker molecule relates to the enzymes used for detection.

### 3.3. Immunosensors

Immunoassays are one of many analytical methods that have been developed and play a significant role for measuring the analytes. These methods have been widely applied in biomedical diagnosis, detection of food contaminants, and environmental analysis. However, further development of new assays with high sensitivity, specificity, and an affordable cost have attracted much attention recently. The biological recognition probes such as antibodies, peptides, and aptamers are conjugated to the nanoparticles and lead to the development of highly sensitive methods. These biomolecules can maintain their conformational orientation as well as interact with the target antigens. The concentration of analytes can be quantified based on the detection of signals generated due to nanoparticles. Amongst several metallic nanoparticles, AuNPs are the most often used labels, which are employed both in immunosensors and DNA sensors [29,30].

### 3.4. DNA Sensors

DNA probes complementary to a specific pathogen DNA sequence and labelled with AuNPs have been widely used in the biosensing of plant pathogens. The presence of DNA sequence as a detection probe provides high specificity to biosensors. DNA biosensors are the analytical tools employed for the sequence specific DNA detection of various pathogens from clinical, food, and environmental samples. Electrochemical DNA biosensors have several advantages due to their affordable cost, sensitivity, and rapid signal generation. The application of AuNPs on the DNA biosensor plays a significant role by the immobilization of DNA on the electrode surface and helps to detect the DNA hybridization events as a label [2]. The use of AuNPs provides better a surface area for the conjugation of the mercaptohexyl group at the 5′-phosphate of the DNA, which enhances the nucleic detection. It has been studied that the surface density of the colloidal Au-modified electrode with ssDNA was 1.0 × 10^14^ molecules/cm^2^ and 10 times higher than on a bare gold electrode [31].

## 4. Pathogen Biosensing

Biosensors have an important role in the timely and rapid detection of several quarantine pathogens of plants, and this could avoid the introduction of exotic pathogens to newer environments. AuNPs have been widely used to label antibodies specific to target pathogens and developed as diagnostic device. 

### 4.1. Bacterial Pathogen Detection

The detection of bacterial pathogens is possible by capturing either the whole bacteria or by targeting the DNA sequence specific to the bacteria. Capturing the DNA probe is usually sensitive and specific to the disease. Colloidal AuNPs were used to label single stranded DNA (ssDNA) probes specific to *Acidovorax avenae* subspecies *citrulli,* the bacterial fruit blotch causing pathogen, and a strip-based DNA sensor was constructed to detect the presence of pathogen rapidly and in on-site settings. Herein, the internal transcribed spacer (ITS) region was used to design the pathogen specific probe. This dipstick method was sensitive enough to detect 4 nM of target DNA qualitatively. Semi-quantitative detection was also possible through analysis of optical density and target DNA concentration. The LOD was found as 0.48 nM [32]. In another example of the detection of *Pseudomonas syringe* pathovars, which causes large scale bacterial diseases in crop plants, AuNPs labelled DNA probes were used in a colorimetric detection of pathogen DNA molecules. Specific primers were designed from conserved N-terminal region of the hrcV gene and probes were then designed with thiol-capping at 5’ or 3’ ends of the probes. The colorimetric detection led to color change from red (non-hybridized AuNP labelled DNA probe) to purple (probe becomes hybridized to target DNA) leading to the identification of pathogen DNA present in the sample [33]. A similar strategy was followed for the detection of soil bacterium *Ralstonia solanacearum,* which causes wilt disease in potatoes. The direct detection of unamplified *R. solanacearum* DNA occurred in a colorimetric AuNP-probe based assay, where the hybridisation of a specific AuNP bound probe with target DNA prevents any aggregation of the AuNPs in the presence of acidic conditions, and thus the colour of the solution remains red indicating the presence of specific *R. solanacearum* DNA. In negative samples, there is no hybridization, and hence AuNPs aggregate and colour change to purple can be detected. The nano-biosensor was found to be rapid, sensitive, and specific towards the detection of *R. solanacearum* directly from soil samples without any DNA amplification steps [34].

In this regard, an electrochemical biosensor based on sandwich immunoassay involving the capture antibody, the detection antibody, AuNPs, and enzyme horseradish peroxidase (HRP) was developed for the detection of quarantine bacterial pathogen *Pantoea stewartii* subspecies *stewartii*-NCPPB 449 (PSS), which is responsible for causing Stewart’s vascular wilt in maize. The high conductivity and surface area properties of AuNPs was utilised to amplify the electrochemical signal and conjugate the HRP-labelled anti-PSS detection antibody, which were then used to detect the presence of bacterial cells bound to the capture antibody. The current response upon the addition of HRP substrate and its subsequent catalytic activity was measured, and a linear relationship was observed with bacteria concentration. The developed biosensing assay was a limit-of detection of 7.8 × 10^3^ cfu/mL higher than the conventional ELISA, and further showed a non-specificity to other selected pathogens [35]. In another study, colloidal gold nanoparticles were used to label monoclonal detection antibodies raised against *Pantoea* subspecies *stewartii*-NCPPB 449 in a strip-based immunoassay based on sandwich ELISA. Different dilutions of pathogens were tested to check the sensitivity of the test and the limit of detection was found to be 1 × 10^5^ cfu/mL in standard and spiked samples with no cross reactivity shown with other pathogens [36].

Surface-enhanced Raman scattering (SERS) is another useful technique that allows highly sensitive and specific detection of molecules of biological or chemical origin. It has been a widely employed technique in the detection of disease-causing pathogens in plants, animals, and humans as well as food-borne pathogens. Using a combination of SERS based methodology and an isothermal DNA amplification technique. Lau et al. developed a multiplex point-of-care system to achieve the detection of *Botrytis cinerea* and *Pseudomonas syringae* along with the fungal pathogen *Fusarium oxysporum* in tomatoes and *Arabidopsis thaliana* [37]. These pathogens widely affect crop species worldwide and hence the timely and rapid detection is of paramount importance in reducing crop losses. AuNPs were tagged with pathogen-specific Raman reporter as well as DNA capture probe to function as SERS nanotags. The recombinase polymerase amplification (RPA) method was used to amplify pathogen-specific amplicons with a 10 nt 5′ overhang and a biotin label at the other end, being amplified from pathogen genomic DNA isolated from the infected plant. Hybridisation of RPA products with SERS nanotags (the 5′ overhang in RPA amplicons and its complementary AuNP bound DNA probe) was then detected after capturing the biotin-RPA-SERS products with streptavidin tagged magnetic beads and exposing them to laser excitation in a portable Raman spectrometer. Distinct peaks were observed for specific pathogens as per the SERS nanotags and RPA amplicon hybridization and the presence of *B. cinerea*, *P. syringae,* and *F. oxysporum f.sp conglutinans* could be detected. The developed system was sensitive enough to detect *B. cinerea* DNA as low as two copies and could easily be multiplexed both in the case of tomatoes and Arabidopsis thaliana systems and was successfully demonstrated in outside laboratory settings.

In yet another study, similar RPA based target DNA amplification was coupled with an AuNP based electrochemical sensing platform to sensitive and specifically detect *P. syringae*, the common plant pathogenic bacterium. In a study by Lau et al., it was found that once the target DNA region was amplified through RPA, it was hybridized with capture DNA probes labelled on to AuNPs and the conjugate was then separated through the binding of biotin tags present in RPA amplicons and streptavidin tagged magnetic beads. The whole complex was heat treated to denature dsDNA and release AuNPs to bring about the electrochemical reduction of Au^+3^ to Au^0^ on a screen-printed carbon electrode and was measured through differential pulse voltammetry (DPV). The more AuNPs are released, more is the amount of target DNA present in the sample and hence pathogen presence could be detected. The total time to carry out the assay was 60 min and the amplification of target DNA through RPA, as well as the detection limit of the electrochemical assay, were more sensitive (100 times each) when compared to end point PCR and gel electrophoresis. Overall, the developed AuNP-electrochemical biosensor was 10,000 times more sensitive and had the capability to detect pathogen presence in the very early stages of infection when tested on *P. syringae* infected *A. thaliana* [23]. In an attempt to make the lateral flow immunoassay (LFIA) biosensor-based detection of potatoes wilt pathogen *R. solanacearum* more effective by reducing the limit-of-detection value, wherein the AuNPs undergo gold enhancement [14]. Table 2 presents the detection of plant pathogens and various sensing methods, as well as their respective LOD.

### 4.2. Detection of Fungal Pathogens

Pathogenic fungal species cause a massive loss in crop quality and yields, posing a threat to the economics of global agricultural sector. It has been reported that approximately 8000 species of fungi and oomycetes are causing diseases in plants and other agricultural crops [47]. Fungi can cause diseases at any stages of plant growth either alone or in association with other kinds of phytopathogens under natural environmental conditions [48]. The most common diseases caused by pathogenic fungi are anthracnose, blight, canker, damping off, dieback, gall, leaf spot, powdery mildew, rust, root rot, scab, and wilt [49]. There are several conventional methods such as polymerase chain reaction (PCR) (nested, multiplex, quantitative, magnetic-capture hybridization PCR techniques) and isothermal amplification methods (LAMP), have been developed for the detection of fungal species. However, these techniques require highly skilled personnel, tedious protocol, and longer time to obtain the results. Recently, with the advent of nanotechnology-based biosensors provide an excellent option for the detection of harmful fungal pathogens. *Phytophthora infestans* is one of the devastating fungi, a causal agent of late blight in potatoes and tomatoes, and a threat to the global agriculture. Therefore, rapid and early detection *P. infestans* is an essential step to contain the disease from further spread. Zhan et al. integrated universal primer mediated asymmetric PCR with AuNPs-based LFA for the visual detection of *P. infestans* [46]. Herein, the asymmetric PCR was performed to produce a large amount of ssDNA and then sandwich hybridization was performed in LFA. In the presence of target DNA, sandwich-type hybridization reactions among the AuNP–probe, target DNA, and capture probe on the test line of LFA, and the distinct red visible line was produced due to the accumulation of AuNPs. The quantification of the LFA was achieved by measuring the signal intensity of the red line and the LOD was found to be 0.1 pg/μL. The detailed schematics of the assay are depicted in Figure 3A.

Black sigatoka is a harmful disease caused by the hemi-biotrophic fungus *Pseudocercospora fijiensis* in banana plantations globally. The detection of this fungal pathogen is important to minimize the losses as well as to prevent the spread to the neighboring cultures. To tackle this pathogen, a highly specific SPR immuno-sensor was developed by Luna –Moreno et al., 2019. The sensor was developed by covalent immobilization of polyclonal antibody (anti-HF; produced against HF1 cell wall protein of *P. fijiensis*) on Au-coated chip via a mixed self-assembled monolayer (SAM) of alkanethiols [45]. The LOD of the SPR immunosensor was found to be 11.7 µg/mL, with a linear response range from 39.1 to 122 µg/mL for the cell wall antigen. The study indicated that there were no matrix effects observed during the analysis of actual leaf banana extracts. Lei et al., 2021, developed an AuNPs-enhanced dynamic microcantilever (MCL) and isothermal recombinase polymerase amplification (RPA) for the detection of *Leptosphaeria maculans*, a virulent phyto-pathogen of oilseed rape [50]. It has been studied that *L. maculans* produces a phytotoxin, i.e., sirodesmin PL and can sustain for a long period under ambient weather conditions. The detection results showed that the sensitivity of the RPA-MCL assay is higher than that of the previously studied fluorescence RPA assay, with a LOD of one copy of *L. maculans* DNA. Furthermore, the rapid and sensitive detection of fungal spores is of great interest due to the potential negative impacts on agriculture as well as public health. Lee et al., 2021, developed a technique for the detection of *Aspergillus niger* spore based on the specific peptides as a recognition probe and the AuNPs as the detection label [51]. The peptides enable rapid binding to the *A. niger* spore resulting in a visible change of colour intensity of the supernatant after sedimentation of the spores. This colorimetric assay displayed a high-sensitivity of −50 spores within <10 min when employed with a smartphone-enabled image analysis application (Figure 3B).

### 4.3. Detection of Viruses

Compared to other types of nanomaterials, AuNPs provide an ideal tool for the virus detection due to numerous reasons, which are already described in the previous sections. The ease of synthesis, surface modifications, stability and biocompatibility, and high absorption coefficient leverages their application in detection platform. The intense red colour can be easily visualized and forms stable bio-conjugates with other biological moiety such as DNA, antibodies, and proteins enabling highly sensitive specific sensing of target analytes. Currently, there are several methods available for the detection of plant virus employing the AuNPs. The methods are based on the colorimetric, fluorescence, or electrochemical, etc.

Cucumber green mottle mosaic virus (CGMMV) causes a severe mosaic symptom of watermelon and cucumber, and can be transmitted via infected cucumber seeds, leaves, and soil. Considering the high transmissibility of the viruses, the early detection is extremely important to control its further spread on the crop fields. Wang et al., 2017, developed a simple and sensitive label-free colorimetric detection method for CGMMV using the unmodified AuNPs as a colorimetric probe [52]. The principle of the assay lies in the binding of RT-PCR target products of CGMMV and species-specific probes, which results in the change in colour after salt (NaCl) induction. Normally, the species-specific probes attach to the surface of AuNPs and thereby increasing their resistance to NaCl-induced aggregation. The developed method did not need any expensive instruments and was capable to detect 30 pg/μL of CGMMV RNA by the naked eye. The assay provided the specificity of 100% with good reproducibility. The localized surface plasmon resonance (LSPR) is another characteristic property of the AuNPs and currently exploited for the development of colorimetric bio-sensing methods. The LSPR generally depends upon the shape size and the surrounding medium of the AuNPs. Razmi et al., employed the LSPR of unmodified AuNPs to detect the tomato yellow leaf curl virus (TYLCV) genome in infected plants [15]. The specific DNA probe that was complimentary against the coat protein of TYLCV was designed and hybridized with the extracted total DNA from the infected sample. The hybridization was performed, cooled to room temperature, and the subsequent addition of the AuNPs indicated the change of colour. The colour change due to the AuNPs suggested the presence of the target virus infection visually and confirmed by the UV-Vis spectroscopy. A similar type of visual colorimetric method has also been developed for Begomovirus in chili and tomato plants with a sensitivity of 500 ag/µL of begomo viral DNA [53]. The comparative screening of chili plants for begomoviral infection by PCR and AuNP assay demonstrated that the AuNP assay (77.7%) was better than the commonly used PCR methods (49.4%). More advanced detection methods have been performed using attenuated total reflection (ATR)-based evanescent wave absorption monitoring LSPR of AuNPs [54]. The binding dynamics of AuNPs has been studied on the amine-functionalized surface refractive index sensor and was developed by monitoring the LSPR absorption peak. The schematics of the detection platform has been represented in Figure 4A. The method was employed for the detection of single-stranded DNA of the chili leaf curl virus with a LOD of 1.0 μg/mL for target viral DNA.

The detection of banana bunchy top virus (BBTV) was achieved through an improved AuNPs based dot immunobinding assay (DIBA), which is rapid and simpler than the conventional ELISA [55]. Herein, the AuNPs were conjugated to the primary antibody and LOD of the DIBA was found to be at sap dilution of 10^−2^. Similarly, grapevine leafroll-associated virus 3 (GLRaV-3) is one of the devastating pathogens causing a significant loss in the yield of grapes. For the widespread control of this virus, on-field analytical methods with a high sensitivity are needed. The application of immuno-chromatographic assay (ICA) or lateral flow assay (LFA) for the detection of pathogens and biomarkers have been widely used. The ICA assay employs the AuNPs and provides an easy interpretation of results in the presence or absence of viruses. The key advantages of the ICA are a short analysis time (10–15 min), an ease in sample preparation, and result interpretation. Byzova et al., 2018, developed an ICA for the rapid detection of GLRaV-3 based on the sandwich immunoassay format [36]. The researcher compared three preparations of AuNPs, (51.0 ± 7.9 nm, 28.3 ± 3.3 nm, and 18.5 ± 3.3 nm) and showed that AuNPs with maximal average diameters of 51.0 ± 7.9 nm provides GLRaV-3 detection for its maximal dilutions. The assay exhibited a sensitivity of 100% and a specificity of 92% in comparison with ELISA, and a sensitivity of 93% and the specificity of 92% as compared to PCR (Figure 3B). The LFA employing AuNPs sometimes suffers the issues of sensitivity and limit of detection. To enhance the sensitivity, there are several signal enhancement methods that have been developed and applied for the detection of plant viruses. Panferov et al. developed a silver enhancement method for the detection of potato leafroll virus (PLRV) [42]. The silver enhancement is based on the reduction of silver ions on the surface of AuNPs and enhances the coloration of AuNPs. This was achieved using a mixture of silver lactate and hydroquinone and the subsequent addition of a chloride-containing buffer to stop the silver reduction. The results suggested that the silver enhanced LFIA was 15 times more sensitive (LOD = 0.2 ng/mL; 15 min) when compared with conventional LFIA (LOD= 3 ng/mL; 10 min). Furthermore, the enhanced LFIA capable of identifying the detected PLRV in leaves’ extracts of infected potato in dilutions higher than ELISA. Another type of study was performed by Razo et al., 2018, by using magnetic nanoparticles (MNPs) and AuNPs for the double enhancement of the LFA [56]. Here, the enrichment of the target was carried out by the specific antibodies conjugated to the magnetic nanoparticles and the signal enhancement was performed by MNPs aggregation through AuNPs. The strategy was employed for the detection of potato virus X (PVX), and the sensitivity of 0.25 ng/mL was achieved and exhibited 32 times more sensitive than the non-enhanced LFA (LOD = 8 ng/mL).

Among various immunoassays, electrochemical immunosensors employing AuNPs enable the development of label-free assays having a shorter analysis time and simplicity over labelled strategies [57,58]. Khater et al., 2019, developed a label free impedimetric biosensor for the detection of tristeza caused by citrus tristeza virus (CTV). The sensing platform was based on the screen-printed carbon electrode (SPCE) modified by AuNPs. The thiolated ssDNA were immobilized on the surface of AuNPs to enhance the electrode conductivity. The hybridization with the target DNA was investigated by EIS measurements in Fe(CN_6_)_4^−^_/Fe(CN_6_)_3^−^_ redox system. The sensor was able to detect the CTV nucleic acids with a linear range of range of 0.1–10 μM in the presence of other non-specific DNAs [59]. Watermelon mosaic virus (WMV) is a major phytopathogen of the family Cucurbitaceae and Leguminosae and induces symptoms of mosaic leaves, decreases of leaf areas, and number of fruits, resulting in a serious reduction of yield. Wang et al., 2019, devised a nicking/polymerization strategy for ultrasensitive electrochemical detection of WMV [60]. The detection platform is based on the exonuclease and polymerase activity of T4 DNA polymerase and Mg^2+^-dependent DNAzyme-assisted and hemin/G-quadruplex DNAzyme-assisted cascade amplification strategies. Briefly, the hybridized DNA of the target WMV sequence, i.e., HP1, and P1 was recognized and nicked by nicking endonuclease. The DNA segments were digested in the 3′→5′ direction and was halted at the 3′-terminal G locus with the exonuclease activity of T4 DNA polymerase. The Mg^2+^-dependent DNAzyme was synthesized by T4 DNA polymerase after the addition of the dNTPs, which hybridized with its substrate sequence immobilized on Au electrode and initiated the cleavage round. Subsequently, the caged G-quadruplex sequence was released and formed hemin/G-quadruplex-based DNAzyme, resulting in the generation of electrochemical signals. The assay showed a linear dynamic range of detection in 50 fM to 1 nM with a LOD of 50 fM.

## 5. Prospects and Possibilities in Forestry

Forests possess multiplex ecological systems, which include not only the population of trees, herbs, and wildlife but also nourishes all life forms along with their integral support systems. Forests provide several products and services to mankind, and the most crucial factor for the socio-economic aspect is their productivity. Forest productivity is the flux of biomass production, carbon storage, and the environmental health of the forest. A disturbance in the forest directly impacts environmental oscillations and causes damaging actions that disturb forest strength and assembly [61]. The first step to combat the issue of disease in natural forests, as well as trees-outside-forests (TOF), is to correctly identify the pathogens and their associated physiology. Common challenges with conventional diagnostic methods, namely time consumption, less accuracy, and a cost effectiveness on a larger scale, therefore allow for emerging low-cost methods to advance the precision and swiftness of plant diagnosis of pathogens out of principal importance. Nanoparticle based biosensors are playing a dynamic part in refining the quality of life and its components through various clinical, ecological, and quality-measured applications across the world. Utilization of several nanomaterials such as metal based, carbon allotropes, polymers, composites, etc. in the form of nanoparticles, nanotubes, nanorods, and nanowires have provided the horizon of faster detection and reproducibility of plant diseases in a much better way. Nano-based biosensors have already proved to be crucial for the diagnosis of several clinical, conservational, and food quality attributes; however, the reports available for plant biosensors are relatively few. Although this approach is being utilized in agricultural crops and plantations in many parts of the world, its efficient utilization in forest areas is highly diverse in natural forests, and hotspots of protected areas are needed, and an integrated approach coupled with institutional management is to be adopted. There are some commercially available nanoparticles based diagnostic sensors from pocket-diagnostic and Loewe, as presented in Figure 5A,B, respectively. In this pocket diagnostic kit, the nanoparticles are used to mix with the extracted genetic material from the plant leaves and provide the positive or negative results in 3–10 min for the respective diseases.

Furthermore, recent progress on the development of smart sensors, such as wearable sensors, where the environmental and plant physiological status can be studied. These provide several benefits, such as the real-time health status of plants inside the firms and in forests. These wearable sensors are small and light enough having stretchability and biocompatibility. Lee et al., developed a nanomaterial based (SWCNT) graphitic electrode and their integration into the plant cell for their real time monitoring of toxic gases. These types of skin-like, flexible, and wearable sensors can also be implemented for the disease detection or early onset of pathogen attack on the firm or in the forest. Apart from this, other advancements include the application of lab-on-drone technology, which combines both sensing and robotic technologies. This will allow sample collection and sample transfer to the LOC microfluidic devices and the results can then be interpreted using a smartphone. The signals from the sensor and the high-quality image processing capacity of the mobile phone could help the farmers, as well as the forest researchers in fighting plant diseases. Research aiming to quantify the influence of changes on forest and associated insect and pathogens is necessary to predict potential disturbance events and associated risks on the forests. This will help forest sustainability and increasing environmental fitness of forests over longer periods.

## Figures and Tables

**Figure 1 sensors-22-01259-f001:**
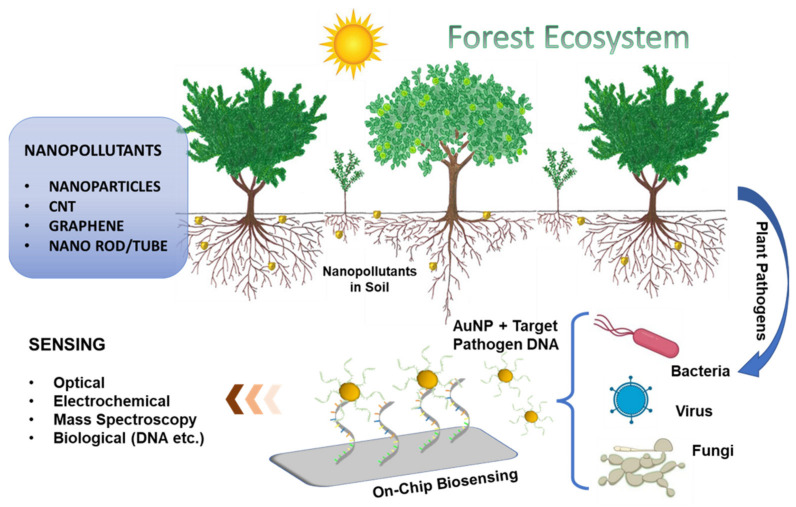
Schematic presentation nanomaterial pollution and use of nanomaterial for plant pathogen detection.

**Figure 2 sensors-22-01259-f002:**
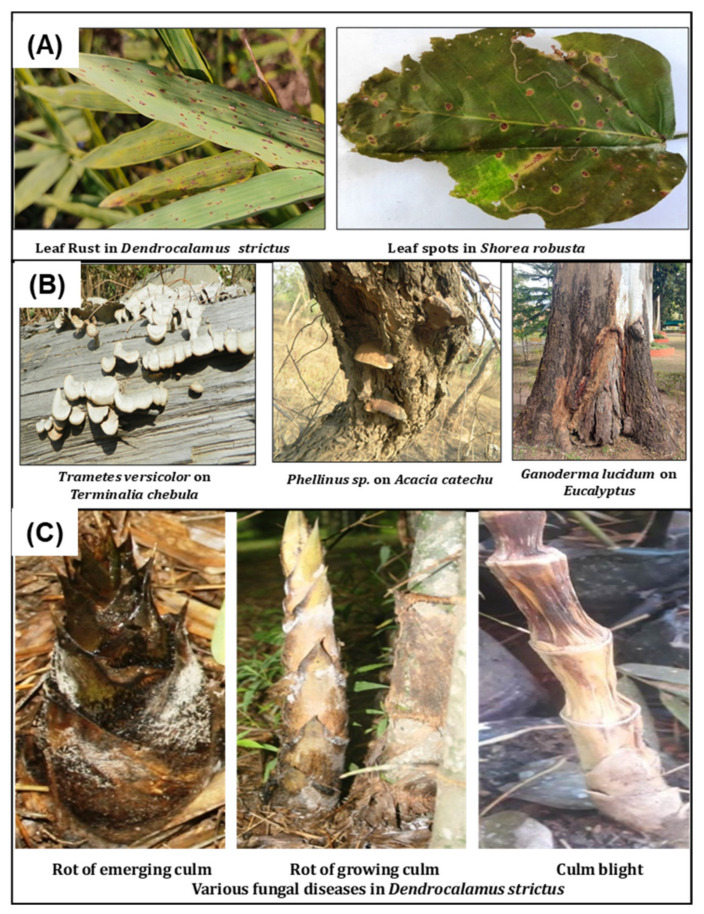
Different Foliar disease: (**A**), stem disease, (**B**) culm disease in bamboo, and (**C**) in forest tree species.

**Figure 3 sensors-22-01259-f003:**
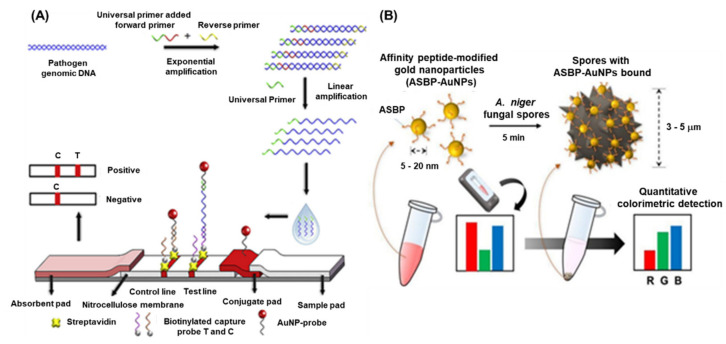
Schematic representation of plant fungi detection by different methods: (**A**) AuNPs based LFA for the detection of *Phytophthora infestans,* [46] (**B**) schematic representation of peptide conjugated AuNPs for the detection of *A. niger* spores. The images are adopted from the references [50].

**Figure 4 sensors-22-01259-f004:**
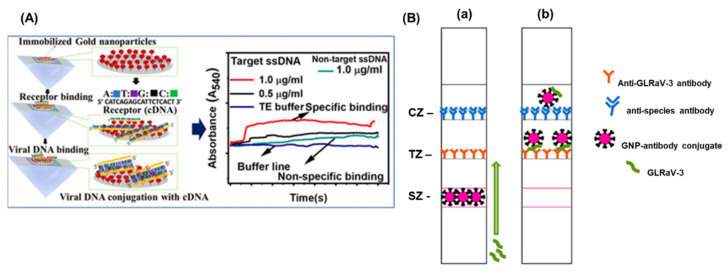
(**A**) Schematic representation of the detection of Chili Leaf Curl Virus using Attenuated Total Reflection-Mediated Localized Surface-Plasmon-Resonance-Based Optical Platform (**B**) Lateral Flow Immunoassay for Rapid Detection of Grapevine Leafroll-Associated Virus. (**a**) is before capturing the GLRaV-3 and (**b**) after binding of GLRaV-3 with antibody. The images are adopted from the references [54,36].

**Figure 5 sensors-22-01259-f005:**
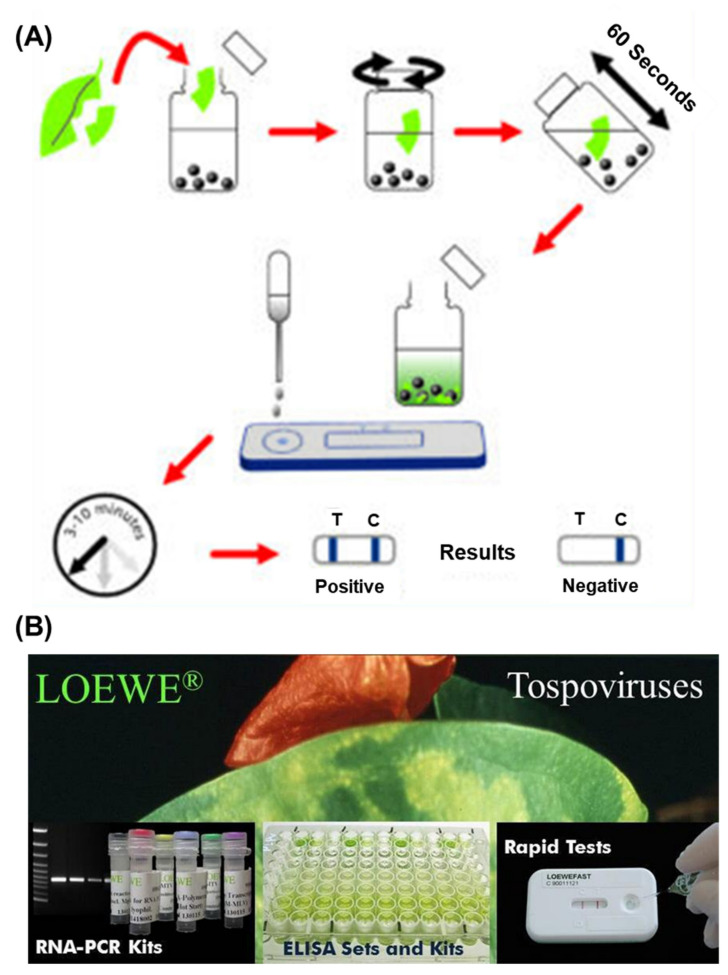
Commercially available rapid diagnostic kit for plant diseases: (**A**) method from pocket diagnostic and (**B**) Loewe RNA-PCR kit for virus. Adopted from www.loewe.com and www.pocketdiagnostic.com, accessed on 4 February 2022.

**Table 1 sensors-22-01259-t001:** Major historical outbreaks of forest tree diseases across the world [17].

Sl. No.	Disease Name	Pathogen	Area and Timeline	Host
1	Dutch Elm Disease	*Ophiostoma-novo-ulmi* (Fungus)	Northwest Europe, 1910s	*Ulmus* sp.
2	Chestnut Blight	*Cryphonectriaparasitica* (Fungus)	USA, Canada, and Asia, 1904	*Castanea sativa*
3	Beech Bark Disease	*Cryptococcus fagisuga* and *Nectria* Fungus (Insect Fungal Complex)	Northern America, 1920–1930s	*Fagus grandifolia*
4	Butternut Canker	*Sirococcus* sp. (Fungus)	North America and Eastern Canada	*Juglans cinerea*
5	Sudden Oak Death	*Phytophthora ramorum* (Fungus)	Oregon, California, and Europe	*Oak* sp.
6	White Pine Blister Rust	*Cronarticumribicola* (Fungus)	Baltic, Russia, 1854	*Pinus parviflora*
7	Jarrah Dieback	*Phytophthora cinnamomi* (Fungus)	Sumatra, Indonesia, 1922	*Eucalyptus marginata*
8	Fire Blight of Pome	*Erwinia amylovora* (Bacterium)	New York, 1780	Pea, apple, and *Rosaceous* spp.
9	Pine Wilt	*Bursaphelenchusxylophilus* (Pine Wood Nematode) spread by *Monochamus* spp. Beetle (Pine Sawyer Beetle)	North America, East Asia, 1940s	*Pine* spp.
10	Scleroderris Canker	*Gremmeniellaabietina*	Canada, 1980s	Coniferous forests
11	Shisham Mortality (i) Fusarium Wilt (ii) Ganoderma Root Rot (iii) Phellinus Root and Butt Rot (iv) Root Knot Nematode	(i) *Fusarium solani* (Fungus) (ii) *Ganoderma lucidum* (Fungus) (iii) *Phellinus gilvus* (Fungus) (iv) *Meloidogyne javanica* (Nematode)	(i) North India (ii) North-Central India (iii) North-Central India (iv) Dehradun, India	*Dalbergia sissoo*
12	Sandal Spike Disease	*Phytoplasma*	Southern India, 1903	*Santalum album*

**Table 2 sensors-22-01259-t002:** Different sensing approaches and nanomaterial used for plant pathogen detection.

No	Plant Disease/Pathogen	Species	Nanomaterial Used	Sensing Method	LOD	Ref.
1	Tomato Yellow Leaf Curl Virus (TYLCV)	Tomato	AuNps with colorimetric nano-biosensing	Localized surface plasmon resonance	5 ng	[15]
2	Cucumber Mosaic Virus (CMV) and Papaya Ring Spot Virus (PRSV)	Papaya	Nanowire based biosensor	Amperometry detection	0.1 mA/mL	[38]
3	Witches’ Broom Disease (*Candidatus Phytoplasma aurantifolia)*	Lime	Quantum dot (QD)-based nano-biosensor	Fluorescence resonance energy transfer (FRET)	5 ca. P. aurantifolia/μL	[39]
4	Odontoglossum Ringspot Virus (ORSV)	Orchid leaves	anodic aluminum oxide (AAO) with AuNPs	Self-assembled monolayer (SAM)	0.345 ng/mL	[40]
5	Bacterial Spot Disease by *Xanthomonas axonopodis*	Solanaceae plant	Fluorescence silica nanoparticles	Fluorescence-linked immunosorbent assay	NA	[41]
6	*Ralstonia solanacearum* (Potato Brown Rot)	Potato	Enlarging AuNPs	Lateral flow immunoassay	3 × 10^4^ cells/mL	[42]
7	Karnal Bunt Disease	Wheat	AuNPs	Surface Plasmone Resonance (SPR)	NA	[43]
8	Powdery Mildew	Rose	Colloidal nanosilver (1.5 nm diameter)	Relative fluorescence units	4.2 μM Ag ions	[44]
9	*Pseudocerocospora fijiensis* Black Sigatoka (Leave Streak Disease)	Banana plants	cell wall protein HF1 of *P. Fijiensis* immobilized onto gold chip	Surface plasmon resonance based immunosensor	11.7 μg/mL,	[45]
10	Late Blight in Potatoes and Tomatoes (caused by Phytophthora Infestans)	Potatoes and Tomatoes	AuNPs	PCR with AuNPs based lateral flow biosensor	0.1 pg/mL range.	[46]
11.	Acidovorax avenae subsp. citrulli	Fruits	Colloidal gold nanoparticles	Dipstick method	0.48 nM of DNA	[32]

## Data Availability

Not applicable.

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
