# Peer review of "Gold Nanoparticles and Plant Pathogens: An Overview and Prospective for Biosensing in Forestry"

_sensors, 2022, doi:10.3390/s22031259_

Round 1

Reviewer 1 Report

a. Since there are various nanomaterials based biosensors it is better to have more discussion on the advantages of using Au nanoparticles over other nanomaterials in detecting plant pathogens in the introduction section besides just the colorimetric detection features.

b. It is great to see that the authors discussed various types of AuNPs based biosensors but most of the representative research discussed in section 2 ‘gold nanoparticles and biosensors’ are related to human diseases rather than plant diseases. Thus, I recommend the authors to replace the representative works in this section with the research cases which are more directly related to plant diseases or pathogens. I am also looking forward to seeing more discussion on the working mechanisms of different type of sensors and the advantages or key roles of using AuNPs in each type of sensors.

c. Section 3 provides a good review of pathogen biosensing based on the types of pathogens. There are parts introducing different types of Au NPs based sensors, which is partially overlapped with Section 2. Therefore, I highly recommend the authors to consier reorganizing the structures of section 2 and section 3.

One idea is to put more introduction of sensor types in section 2, and focus more on how various types of sensors are applied in the detection of various pathogens.

Another idea would be having a thorough introduction of different types of pathogens as section 2, then in the following section putting representative works in detection of pathogens into corresponding subsections of AuNPs biosensors by sensor type to replace the current human disease related Au NPs biosensor examples.

d. In section 4, I am very curious to see the authors to discuss more on why nano-based biosensors haven’t been widely used in plant diseases, what are the challenges, and more input on perspective directions to solve the current problems.

Author Response

The authors like to thank you for your valuable time and constructive suggestion to further improve the review article. We tried to incorporate all the suggestions and added further details as required. 

Reviewer 2 Report

In general, this review it's well written, complete, with scientific soundness, and well organized. The presented information is clear. My recommendation is to accept the article for publication.

Author Response

The authors like to thank you for your consideration of the review article. 

Reviewer 3 Report

This manuscript presents an interesting review about the gold nanoparticles and plant pathogens for biosensing in forestry. This review includes nanomaterial based biosensing methods of disease detection. In addition, this review considers the requirements for a reliable, fast, and cost-effective testing method for monitoring plants in forest and plants during their early stages of growth. This manuscript can be improved considering the following comments:

1.-English grammar and style of all the sections of manuscript must be improved.

2.-Introduction should add the importance and advantages of the application of nanomaterials for biosensing in forestry. In addition, this section should incorporate better description of the investigations of nanomaterials for biosensing in forestry.

3.-This manuscript has very few figures. Authors should add more figures for a better understanding of the different investigations about the gold nanoparticles, biosensors, and pathogen biosensing.

4.-Second and third sections require more description about the main performance parameters, advantages, and limitations of the gold nanoparticles, biosensors, and pathogen biosensing. This description should include schematic view and figures of the nanomaterials and biosensors.

5.-The quality of figures 2 and 3 must be improved.

6.-More discussion about the performance and reliability of biosensors and pathogen biosensing should be considered.

7.-Authors could include a section about the different techniques of signal processing used for the biosensors and pathogen biosensing.

8.-Section of limitations or challenges about the gold nanoparticles, biosensors, and pathogen biosensing should be included.

9.-References must be checked. The format of references must be written using the Sensors format.

Author Response

(The authors gave the same response as above.)

Round 2

Reviewer 1 Report

The revised version covered all my comments very well. Only one tiny suggestion is to revise the subtitle of section 3 to "gold nanoparticle based biosensors".

Author Response

The authors Like to thank reviewers for their valuable time and suggestions for improvement of the MS. We also thank you for acceptance of the MS after revision.  The subtitle of section 3 is revised as mentioned. 

Reviewer 3 Report

Authors have improved their manuscript based on the comments of reviewer. This revised manuscript can be accepted for publication in Sensors.

Author Response

The authors Like to thank reviewers for their valuable time and suggestions for improvement of the MS. We also thank you for acceptance of the MS after revision.